

# Identification of PIMREG as a novel prognostic signature in breast cancer via integrated bioinformatics analysis and experimental validation

Wenjing Zhao[1,*], Yuanjin Chang[2,*], Zhaoye Wu[2], Xiaofan Jiang[2], Yong Li[1], Ruijin Xie[2], Deyuan Fu[1], Chenyu Sun[3,4] and Ju Gao[1]

[1] Clinical Medical College, Yangzhou University, Yangzhou, Jiangsu, China
[2] School of Medicine, Jiangnan College, WuXi, JiangSu, China
[3] Department of General Surgery, The second Affiliated Hospital of Anhui Medical University, Anhui, China
[4] Department of Medicine, AMITA Health Saint Joseph Hospital, Chicago, IL, USA
[*] These authors contributed equally to this work.

## ABSTRACT

**Background**. Phosphatidylinositol binding clathrin assembly protein interacting mitotic regulator (PIMREG) expression is upregulated in a variety of cancers. However, its potential role in breast cancer (BC) remains uncertain.

**Methods**. The Cancer Genome Atlas (TCGA) and Gene Expression Omnibus (GEO) databases were used to gather relevant information. The expression of PIMREG and its clinical implication in BC were assessed by using Wilcoxon rank-sum test. The prognostic value of PIMREG in BC was evaluated through the Cox regression model and nomogram, and visualized by Kaplan-Meier survival curves. Genes/proteins that interact with PIMREG in BC were also identified through GeneMANIA and MaxLink. Gene set enrichment analysis (GSEA) was then performed. The correlations of the immune cell infiltration and immune checkpoints with the expression of PIMREG in BC were explored via TIMER, TISIDB, and GEPIA. Potential drugs that interact with PIMREG in BC were explored via Q-omic. The siRNA transfection, CCK-8, and transwell migration assay were conducted to explore the function of PIMREG in cell proliferation and migration.

**Results**. PIMREG expression was significantly higher in infiltrating ductal carcinoma, estrogen receptor negative BC, and progestin receptor negative BC. High expression of PIMREG was associated with poor overall survival, disease-specific survival, and progression-free interval. A nomogram based on PIMREG was developed with a satisfactory prognostic value. PIMREG also had a high diagnostic ability, with an area under the curve of 0.940. Its correlations with several immunomodulators were also observed. Immune checkpoint CTLA-4 was significantly positively associated with PIMREG. HDAC2 was found as a potentially critical link between PIMREG and BRCA1/2. In addition, PIMREG knockdown could inhibit cell proliferation and migration in BC.

**Conclusions**. The high expression of PIMREG is associated with poor prognosis and immune checkpoints in BC. HDAC2 may be a critical link between PIMREG and BRCA1/2, potentially a therapeutic target.

Corresponding authors
Chenyu Sun, drsunchenyu@yeah.net
Ju Gao, gaoju_003@163.com

## INTRODUCTION

Breast cancer (BC) is the most commonly diagnosed malignant tumor among females worldwide. It is the leading cause of cancer-related fatalities, accounting for 6.9% of all cancer diagnoses (*Breastcancer.org, 2023*). It imposes substantial disease and economic burdens on both developed and developing countries (*Najafi et al., 2019*; *Sung et al., 2021*). Despite the fact that early BC screening and identification are critical for reducing morbidity and mortality by locoregional surgery, conventional chemotherapy, and radiotherapy, the recurrence rate of BC remains high, leading to an elevated risk of death. (*Hamann et al., 2019*; *Mayor, 2015*). The prediction of clinical prognosis in BC largely depends on our understanding of the clinical, pathological, and molecular characteristics of BC. Still, the underlying etiology of BC aggressiveness is yet unknown. Identifying potential diagnostic and prognostic biomarkers of BC would help both detect BC patients and understand BC etiology (*Arnedos et al., 2019*; *Duffy et al., 2015*).

Phosphatidylinositol binding clathrin assembly protein interacting mitotic regulator (PIMREG), also known as FAM64A, was discovered to be a clathrin assembly lymphoid myeloid leukemia gene (CALM) interactor. It is located at 17p13.2 and plays the key role in modulating the subcellular localization of the leukemogenic fusion protein CALM/AF10 (*Archangelo et al., 2008*). It has been shown that PIMREG controls the transition from metaphase to anaphase of the cell's mitotic cycle, so it could be a marker for the proliferation and tumorigenesis of various cancer kinds (*Barbutti et al., 2016*; *Zhao et al., 2008*). Recent studies revealed that PIMREG is upregulated in many different types of cancer, such as prostate cancer, lung cancer, gastric cancer, esophageal cancer, colorectal cancer, and most importantly, BC (*Jiang et al., 2021*; *Zhou et al., 2021*; *Zhu et al., 2021*). Also, elevated PIMREG expression is associated with a worse prognosis among patients with lung, liver, pancreatic, and other types of cancer. However, the underlying mechanism of PIMREG in BC, its prognostic value in different types of BC, and its diagnostic value in BC are yet to be further explored. Moreover, the previous research indicated that PIMREG overexpression significantly activates NF-$\kappa$B signaling and promotes breast cancer aggressiveness, and the immune infiltrates associated with the upregulation of PIMREG in BC require further investigation (*Jiang et al., 2019*), and potential therapies targeted to PIMREG in BC need to be developed.

In this study, we combined bioinformatics analysis and cell biology experiments to investigate the roles and clinical values of PIMREG in BC. We first explored the prognostic and diagnostic significance of PIMREG using data extracted from the Cancer Genome Atlas (TCGA) database. Then, we investigated the role of PIMREG in BC pathogenesis through Gene set enrichment analysis (GSEA) and network analysis. Furthermore, we examined the correlation of PIMREG with immune checkpoints and the level of immune infiltration. We also explored drugs that potentially respond to overexpression of PIMREG in BC, as they might offer novel immunological perspectives to understand the mechanism of tumor

progression and provide useful evidence for future studies regarding the immunological and pharmacological therapies for BC patients. Finally, experimental validation was conducted to verify the expression of PIMREG in BC and explore its function on cell proliferation and migration in BC. This study aimed to identify a prognostic signature and provide future cancer therapeutics in breast cancer.

## METHODS

### Analysis of PIMREG expression level

The gene expression data of 1109 BC tissues and 113 adjacent normal tissues were obtained from the Breast Invasive Carcinoma Project (TCGA-BRCA) of the TCGA database. The clinicopathological characteristics of the patient's data were summarized. Wilcoxon rank-sum test was used to assess the expression level of PIMREG between BC and adjacent normal tissue; visualization was performed using the 'ggplot2' package in R software v3.6.3 (*R Core Team, 2020*; *Ding et al., 2022*). Further analysis was conducted on information obtained from 112 pairs of breast cancer tissue samples and adjacent normal breast tissues. Moreover, subgroup analyses were conducted based on T, N, M stage, pathologic stage, PR status, HER-2 status, and ER status. Statistical significance was considered when $p < 0.05$. Findings were verified through the GSE42568 data set, which including104 BC tissues and 17 normal tissues. Data obtained from Gene Expression Omnibus (GEO) database set aimed to analyze the expression difference of PIMREG between BC tissues and normal breast tissue (*Clarke et al., 2013*; *Zhou et al., 2020*). All figures were visualized using the R package ggplot2. To evaluate and verify the expression of PIMREG in BC, images of immunohistochemical (IHC) staining for the PIMREG protein obtained from the Human Protein Atlas (HPA) (http://www.proteinatlas.org/) were used (*Pontén et al., 2011*; *Zhu et al., 2022*).

### Prognosis analysis of PIMREG expression within BC

Using RNAseq data with Log2 transformed TPM (transcripts per million reads) of TCGA-BRCA, the prognostic significance of PIMREG expression was first investigated. To compare the survival difference, as well as overall survival (OS), disease-specific survival (DSS), and progression-free interval (PFI), between high PIMREG and low PIMREG expression groups, we applied the Kaplan–Meier (KM) survival analysis with the log-rank test. Log-rank tests and univariate Cox proportional-hazards regression were performed to create the KM curves, which included *p*-values and hazard ratio (HR) with a 95% confidence interval (CI). The KM curve of OS based on the GEO database (GSE42568) was also plotted with the same method. R software with 'survminer' and 'survival' packages was used to perform these calculations. Furthermore, using the "plotROC" package, the diagnostic property was verified through receiver operating characteristic (ROC) analysis of BC. Meanwhile, in order to explore and further clarify the risk factors in patients with BC, the univariate analysis of the correlation between clinicopathological characteristics and OS in BC was also performed. A forest plot was developed to visualize these univariate and multivariate analyses. Based on overexpressed PIMREG and other potential predictors, a predictive model based on TCGA-BRCA data was also developed to predict the mortality

risk (*Balachandran et al., 2015*; *Iasonos et al., 2008*; *Liu et al., 2018*). A nomogram using 'rms' and 'survival' R packages was constructed to create a graphical representation of potential predicting factors for calculating mortality risk for an individual patient and predicting the 1-, 3-, and 5-year overall survival. All figures were visualized using the R package 'ggplot2'.

## GSEA

Based on transcriptional sequences obtained from TCGA, GSEA was used to identify gene sets and pathways associated with PIMREG. Gene expressions were classified into high-expression and low-expression groups. To assess the potential functions of these groups, we conducted a comparative analysis using GSEA *via* the Broad Institute website (https://www.gsea-msigdb.org/gsea/index.jsp), utilizing the R package cluster profiler (*Subramanian et al., 2005*; *Yu et al., 2012*). The results of this analysis were visually presented using the R package 'ggplot2'.

## Explore other genes/proteins that interacted with PIMREG

The protein-protein interaction (PPI) networks of the PIMREG were constructed *via* GeneMANIA (http://www.genemania.org) (*Vlasblom et al., 2015*). In addition, the MaxLink web server (https://funcoup5.scilifelab.se/maxlink/) was conducted to find and rank other cancer gene candidates by their connectivity to the PIMREG and the two most commonly expressed genes for BC, *i.e.*, BRCA1 and BRCA2 (*Guala, Sjölund & Sonnhammer, 2014*).

## Infiltration of immune cells

We utilized TIMER2.0, a web-based tool for comprehensive molecular characterization of tumor-immune interactions, to examine the infiltration of different immune cells and their clinical effects (*Li et al., 2017*). Using the Gene module in TIMER2.0, we created plots to explore the correlation between the expression of PIMREG members and immune infiltration level in BC. A significant correlation is set for the cutoff value of Cor >0.2 and p<0.05 (*Li et al., 2016*; *Liu et al., 2021*; *Zhong et al., 2021*).

To investigate interactions and relationships between the immune system and PIMREG, we employed the TISIDB database (http://cis.hku.hk/TISIDB). TISIDB is an online platform for evaluating immune system and tumor interactions, including almost 1,000 reported immune-related anti-tumor genes, etc., and immunological data aggregated from seven public databases (*Ding et al., 2021*; *He et al., 2021*; *Xu et al., 2021*). This study used TISIDB to explore highly associated immunomodulators with PIMREG in BC.

## Relationship between PIMREG and immune checkpoints in BC

To investigate the potential oncogenic role of PIMREG in breast cancer (BC), we assessed its associations with immune checkpoints using TIMER and GEPIA. In this study, we focused on the immune checkpoints programmed death 1 (PD1)/PD1-L1 and cytotoxic T-lymphocyte-associated protein 4 (CTLA-4) as they are well-recognized critical checkpoints in tumor immune evasion (*De Velasco et al., 2017*; *Li et al., 2017*; *Lou et al., 2021*; *Xu et al., 2018*).

**Table 1   Clinical information of patients involved in this study.**

| Variables | |
|---|---|
| No of patients | 5 |
| **Age (years, mean)** | 49.2 ± 12.498 |
| **Sex (%)** | |
| Female | 5 (100%) |
| Male | 0 |
| **Tumor size** | |
| ≤2.0 cm | 1 (20%) |
| >2.0 cm | 4 (80%) |
| **Lymph node metastasis (%)** | |
| Yes | 3 (60%) |
| No | 2 (40%) |
| **Menopause (%)** | |
| Yes | 3 (60%) |
| No | 2 (40%) |
| **Stage/grade (%)** | |
| I∼II | 3 (60%) |
| III∼IV | 2 (40%) |
| Unknown | 0 |

## Investigate potential drugs that demonstrate interaction with PIMREG in BC

Q-omics (version 1.0) was employed to investigate drugs that exhibit a potentially favorable response to PIMREG. The score, developed by Q-omics, represents the log-fold change in drug response between samples with high PIMREG expression and those with low PIMREG expression. This scoring system was utilized to identify drugs demonstrating the strongest response to high PIMREG expression (*Jeong et al., 2020*; *Lee et al., 2021*).

## Human tissue samples collection

We collected five surgically removed breast cancer (BC) tissues and their matched adjacent normal tissues from Northern Jiangsu People's Hospital, affiliated with Yangzhou University, between October 2022 and February 2023. The clinical information of the patients is provided in Table 1. All participants in this study provided written informed consent, and the study protocol was approved by the ethics committee of Northern Jiangsu People's Hospital, affiliated with Yangzhou University (Approval ID: 2022KY316).

## Cell lines

The human mammary epithelial cell line MCF-10A (#CL-0525) and the human breast cancer cell line MDA-MB-231 (#CL-0150B) were obtained from Procell (Wuhan, China). MCF-10A cells were cultured in DMEM/F12 (Gibco, CA, USA) supplemented with 5% horse serum, 0.5 $\mu$g/ml hydrocortisone, 20 ng/ml human epidermal growth factor, 10 $\mu$g/ml insulin, and 1% penicillin/streptomycin. MDA-MB-231 cells were cultured in Dulbecco's modified Eagle's medium (DMEM, Hyclone, UT, USA), supplemented with

10% fetal bovine serum and 1% penicillin/streptomycin. Before further experiments, these cells were incubated in a humidified atmosphere containing 5% $CO_2$ at 37 °C.

## Quantitative real-time reverse transcription PCR (qRT-PCR)

To assess the mRNA levels in cells, quantitative real-time polymerase chain reaction (qRT-PCR) was performed following a previously published protocol (*Xie et al., 2022*). Total RNA was extracted from the cells using the TRizol reagent (Takara, Shiga, Japan). Subsequently, the PrimeScriptTM RT reagent kit (Takara, Shiga, Japan) was utilized to synthesize single-stranded cDNA from the RNA, following the manufacturer's instructions. The $2 - \Delta\Delta Ct$ value method, which involves determining the cycle at which the fluorescence level reaches the threshold limit set by the qPCR system, was employed to calculate the relative gene expression in the target and reference samples (*Rao et al., 2013*). This study used it to calculate the relative mRNA expression of PIMREG or HDAC2. Glyceraldehyde 3-phosphate dehydrogenase (GAPDH) was used as an endogenous control. The sequences of RT-PCR primers were used as follows: PIMREG: forward 5′-CCTGGAAACGCCTGGAAAC-3′and reverse 5′-CAAAGCACTCTTAGCTGAGCG-3′; HDAC2: forward 5'- ATGGCGTACAGTCAAGGAGG-3' and reverse 5'-TGCGGATTCTATGAGGCTTCA-3'; GAPDH: forward 5′-ATTCCACCCATGGCAAATTC-3′and reverse 5′-TGGGATTTCCATTGATGACAAG.

## Western blot assay

To examine the protein expression of PIMREG in cells and BC tissues, a western blot assay was performed based on a previously published protocol (*Xie et al., 2022*). Briefly, the cells or BC tissues were lysed using RIPA reagent (Beyotime, Shanghai, China) and boiled for 10 min. Subsequently, the total proteins were separated by 10% sodium dodecyl sulfate-polyacrylamide gel electrophoresis (SDS-PAGE) and transferred onto PVDF membranes (Millipore, Billerica, MA, USA). The membranes were then blocked with 5% nonfat milk diluted with Tris-buffered saline-Tween (TBST) at room temperature for 1 h and incubated overnight at 4 °C with the primary antibody: anti-PIMREG (1:500; abs152589; Absin Bioscience, Inc., Shanghai, China). The next day, the membranes were incubated with the HRP-conjugated secondary antibodies at room temperature for 1 h. The Western blots were visualized using an ECL detection kit (Proteintech Group, Inc., Rosemont, IL, USA) for chemiluminescence. The density of the protein bands was quantified using Image J software (Pierce, Rockford, IL, USA), and the values was normalized to $\beta$-Actin.

## Immunohistochemistry assay

The immunohistochemistry (IHC) assay followed a previously published protocol (*Jin et al., 2018*). Briefly, paraffin-embedded breast tissue slides with a thickness of 5 $\mu$m were incubated overnight at 4 °C with anti-PIMREG (1:50; abs152589; absin, Shanghai, China). Subsequently, the slides were incubated with the secondary antibody at room temperature for 1 h. Then, the slides were incubated in diaminobenzidine (DAB) and counterstained with hematoxylin. Images of the slides were captured under a microscope.

## Transfection assay

To knockdown the expression of the PIMREG, the siRNA transfection assay was conducted based on the previously published paper (*Yao et al., 2019*). Briefly, cells were seeded into a 6-well plate and transfected with Lipo8000™ Transfection Reagent (#C0533, Beyotime, Shanghai, China) following the manufacturer's instructions. After 48 h of siRNA transfection, the knockdown efficiency was confirmed by conducting a western blot assay or qRT-PCR. The siRNA targeting human PIMREG or HDAC2 mRNA was designed by Genepharma Technologies Inc (Shanghai, China), and the siRNA sequences were used as follows: siPIMREG: 5′-UCCUGGAAACGCCUGGAAATT-3′, si-HDAC2: 5′-GGAUUACAUCAUGCU AAGA-3′.

## Colony formation assay

To investigate the impact of HDAC2 on breast cancer, a colony formation assay was performed following a previously published protocol (*Kim & Ma, 2018*). Briefly, MDA-MB-231 cells were transfected with si-HDAC2 and subsequently seeded in 6-well plates at a density of 1000 cells per well. The cells were then incubated for 7 days to allow colony formation. Afterward, the colonies were fixed with 4% paraformaldehyde for 30 min and stained with 0.5% crystal violet for 30 min. The number of colonies was quantified using ImageJ software.

## Cell counting kit-8 (CCK-8) assay

To assess the cell proliferation of MDA-MB-231 cells transfected with siRNA-PIMREG, a CCK-8 assay was performed following a previously published protocol (*Xie et al., 2022*). Briefly, after transfection with siRNA-PIMREG, MDA-MB-231 cells were seeded in 96-well plates at a volume of 100 $\mu$L/well, with 5000 cells per well. The cells were then incubated for 0 h, 24 h, 48 h, and 72 h. Subsequently, 10 $\mu$L of CCK8 solution (Beyotime, Shanghai, China) was added to each well, and the plate was further incubated for 30 min. Finally, the cell proliferation was measured by recording the absorbance at 450 nm using a microplate reader.

## Transwell migration assay

To assess the migration ability of cells transfected with siRNA-PIMREG, a transwell migration assay was conducted following a previously published protocol (*Yao et al., 2019*). Briefly, after transfection with siRNA-PIMREG, $5 \times 10^4$ cells were seeded in the upper chamber with an 8 $\mu$m pore size (Corning Costar). The lower chamber was filled with DMEM containing 10% FBS to stimulate cell migration. After incubation for 12 h, the migrated cells were fixed with 4% paraformaldehyde for 30 min and stained with 0.1% crystal violet for 30 min. Images of the migrated cells were captured under a microscope.

## Statistical analysis

The bioinformatics analysis in this study was conducted using the online database mentioned above or the R software. The experimental results are presented as mean ± standard deviation (SD), and the differences between the two groups were compared using the Student's *t*-test. Each experiment was repeated three times. Statistical significance was defined as a *P* value <0.05 or a log-rank *P* value <0.05.

## RESULTS

### Correlation analysis of PIMREG expressions in BC

Among the data downloaded from TCGA, a total of 1109 samples were available. However, the demographic and clinical data were only accessible for 1083 patients with breast cancer (Table 2). The analysis revealed significant associations between the expression of PIMREG and various clinicopathological parameters. These parameters include race ($P < 0.001$), T stage ($P < 0.001$), pathologic stage ($P = 0.002$), histological type ($P < 0.001$), PR status ($P < 0.001$), ER status ($P < 0.001$), and PAM50 status ($P < 0.001$) in breast cancer patients. These findings suggest that PIMREG may serve as a promising novel biomarker in the context of breast cancer.

### Upregulation of PIMREG in BC

The analysis of mRNA expression data from the TCGA database revealed a significantly higher average expression level of PIMREG in BC tissues compared to normal tissues ($P < 0.001$) (Fig. 1A). This finding was further confirmed by analyzing matched tumors and surrounding samples, which demonstrated higher expression of PIMREG in BC tumor tissues (Fig. 1B). Consistent results were obtained from the expression difference analysis based on the GEO database ($P < 0.001$) (Fig. 1C). Subsequent analysis showed a significant difference in PIMREG expression between pathologic stage I and stage II, with higher expression observed in stage II ($P < 0.001$) (Fig. 2A). Moreover, more advanced T stage was associated with higher expression of PIMREG ($P < 0.001$) (Fig. 2B). However, there was no association between the M and N stages with PIMREG expression (Figs. 2C–2D). Regarding hormone receptor status, PIMREG expression was found to be elevated in estrogen receptor (ER) negative BC tissues compared to ER-positive tissues, and it was even more increased in progestin receptor (PR) negative BC tissues compared to PR-positive tissues ($P < 0.001$) (Figs. 2E–2F). However, no statistically significant difference was observed between human epidermal growth factor receptor 2 (HER2) positive and HER2-negative BC tissues (Fig. 2G). Additionally, PIMREG expression was significantly higher in infiltrating ductal carcinoma compared to infiltrating lobular carcinoma ($P < 0.001$) (Fig. 2H). To further support these findings, immunohistochemistry (IHC) images obtained from datasets of the Human Protein Atlas (HPA) were used to demonstrate PIMREG protein expression in BC and breast tissues. The intensity of PIMREG staining was weak in normal breast tissue but moderate in BC tissues, confirming the elevated expression of PIMREG in BC compared to normal breast tissue (Fig. 3). Overall, these results indicate that PIMREG is likely overexpressed in breast cancer.

### Prognostic property of PIMREG in BC

The TCGA database was used to assess the impact of the upregulation of PIMREG on BC prognosis. Its upregulation was revealed to be associated with worse OS (HR: 1.48; 95%CI [1.17–2.03]; $P = 0.018$) and DSS (HR: 1.78; 95%CI [1.15–2.76]; $P = 0.009$), as well as shorter PFI (HR: 1.64; 95%CI [1.18–2.28]; $P = 0.003$) in patients with BC (Figs. 4A–4C). In addition, receiver operating characteristic (ROC) analysis was also performed with an area under the curve (AUC) value of 0.940 (95%CI [0.919–0.961]), indicating a high

**Table 2  Correlation between PIMREG and clinicopathological characteristics from TCGA-BRCA dataset.**

| Characteristic | Low expression of PIMREG | High expression of PIMREG | p |
|---|---|---|---|
| n | 541 | 542 | |
| T stage, n (%) | | | <0.001 |
| T1 | 167 (15.5%) | 110 (10.2%) | |
| T2 | 283 (26.2%) | 346 (32%) | |
| T3 | 75 (6.9%) | 64 (5.9%) | |
| T4 | 15 (1.4%) | 20 (1.9%) | |
| N stage, n (%) | | | 0.135 |
| N0 | 259 (24.3%) | 255 (24%) | |
| N1 | 171 (16.1%) | 187 (17.6%) | |
| N2 | 54 (5.1%) | 62 (5.8%) | |
| N3 | 47 (4.4%) | 29 (2.7%) | |
| M stage, n (%) | | | 0.293 |
| M0 | 446 (48.4%) | 456 (49.5%) | |
| M1 | 7 (0.8%) | 13 (1.4%) | |
| Pathologic stage, n (%) | | | 0.002 |
| Stage I | 109 (10.3%) | 72 (6.8%) | |
| Stage II | 283 (26.7%) | 336 (31.7%) | |
| Stage III | 131 (12.4%) | 111 (10.5%) | |
| Stage IV | 7 (0.7%) | 11 (1%) | |
| Race, n (%) | | | <0.001 |
| Asian | 24 (2.4%) | 36 (3.6%) | |
| Black or African American | 59 (5.9%) | 122 (12.3%) | |
| White | 420 (42.3%) | 333 (33.5%) | |
| Age, n (%) | | | 0.030 |
| <=60 | 282 (26%) | 319 (29.5%) | |
| >60 | 259 (23.9%) | 223 (20.6%) | |
| Histological type, n (%) | | | <0.001 |
| Infiltrating Ductal Carcinoma | 335 (34.3%) | 437 (44.7%) | |
| Infiltrating Lobular Carcinoma | 152 (15.6%) | 53 (5.4%) | |
| PR status, n (%) | | | <0.001 |
| Negative | 102 (9.9%) | 240 (23.2%) | |
| Indeterminate | 2 (0.2%) | 2 (0.2%) | |
| Positive | 412 (39.8%) | 276 (26.7%) | |
| ER status, n (%) | | | <0.001 |
| Negative | 50 (4.8%) | 190 (18.4%) | |
| Indeterminate | 0 (0%) | 2 (0.2%) | |
| Positive | 466 (45%) | 327 (31.6%) | |
| HER2 status, n (%) | | | 0.550 |
| Negative | 287 (39.5%) | 271 (37.3%) | |
| Indeterminate | 7 (1%) | 5 (0.7%) | |
| Positive | 74 (10.2%) | 83 (11.4%) | |

**Table 2** (*continued*)

| Characteristic | Low expression of PIMREG | High expression of PIMREG | *p* |
|---|---|---|---|
| Menopause status, n (%) | | | 0.709 |
| Pre | 119 (12.2%) | 110 (11.3%) | |
| Peri | 18 (1.9%) | 22 (2.3%) | |
| Post | 354 (36.4%) | 349 (35.9%) | |
| PAM50, n (%) | | | <0.001 |
| Normal | 31 (2.9%) | 9 (0.8%) | |
| Luminal A | 416 (38.4%) | 146 (13.5%) | |
| Luminal B | 55 (5.1%) | 149 (13.8%) | |
| Her2 | 28 (2.6%) | 54 (5%) | |
| Basal | 11 (1%) | 184 (17%) | |
| Anatomic neoplasm subdivisions, n (%) | | | 0.412 |
| Left | 274 (25.3%) | 289 (26.7%) | |
| Right | 267 (24.7%) | 253 (23.4%) | |
| radiation therapy, n (%) | | | 0.501 |
| No | 230 (23.3%) | 204 (20.7%) | |
| Yes | 280 (28.4%) | 273 (27.7%) | |
| Age, median (IQR) | 60 (49, 68) | 57 (48, 66.75) | 0.085 |

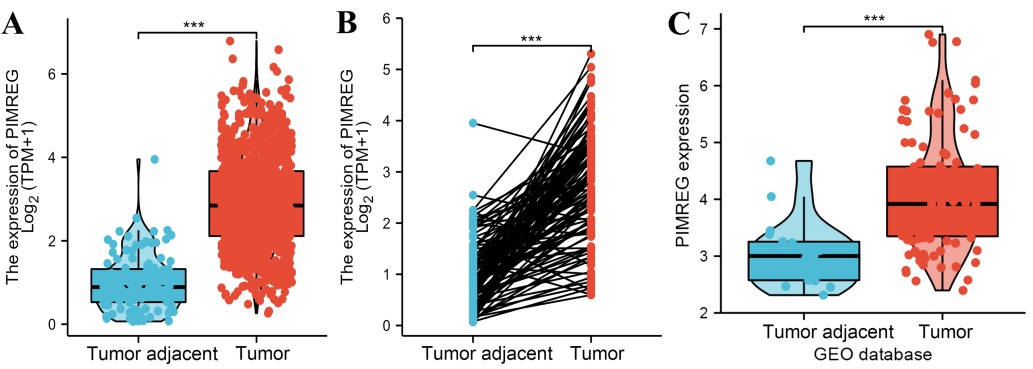

**Figure 1** **The level of PIMREG expression in breast cancer tissue *vs* normal tissues (sample size = 1109 from TCGA database, 104 from GEO database).** (A) Comparison of expression level of PIMREG between breast tissues and normal breast sample based on TCGA database. (B) The expression level of PIMREG in normal breast sample and matched breast cancer tissues based on TCGA. (C) Comparison of expression level of PIMREG between samples from breast cancer and samples from normal breast based on GEO database (GSE42568). Asterisks (***) indicate $p < 0.001$.

prognostic value of PIMREG (Fig. 4D). Further analysis of subgroups found that the upregulation of PIMREG was associated with worse OS in infiltrating lobular carcinoma with HR of 2.79 (95%CI [1.13–6.75]; $P = 0.026$), but not in infiltrating ductal carcinoma. (Figs. 5A–5B) Based on the GEO database, the higher expression of PIMREG was also found to be associated with worse OS in BC (HR: 2.28; 95%CI [1.13–4.59]; $P = 0.021$). (Fig. 5C) As PAM50 classifies BC into five molecular intrinsic subtypes that differ in their

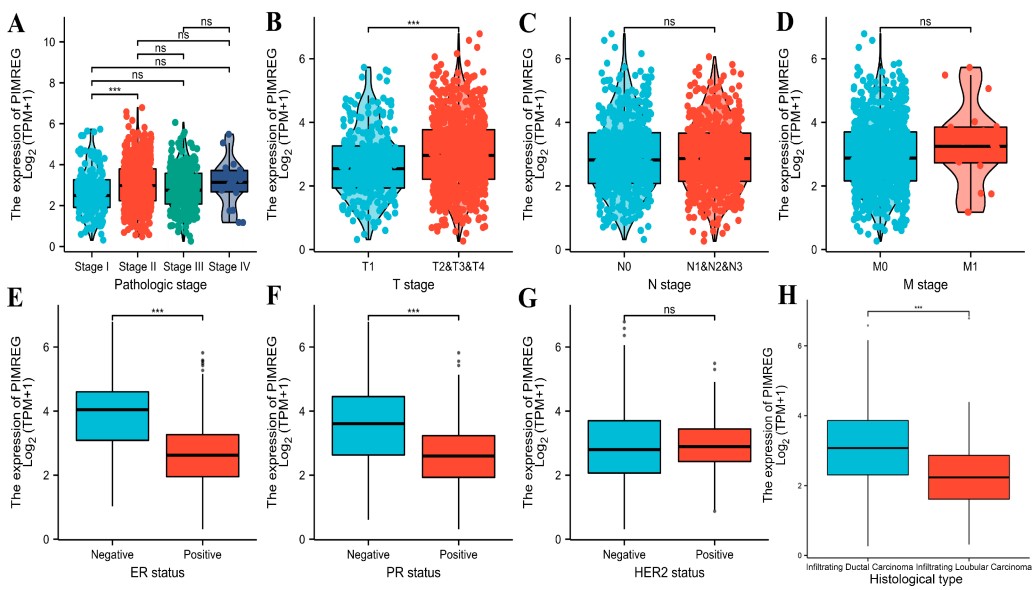

**Figure 2** **Correlation between the expression of PIMREG and certain clinicopathological factors (sample size = 1109 from TCGA database).** (A) Clinical stage. (B) T classification. (C) N classification. (D) M classification. (E) ER status. (F) PR status. (G) HER2 status. (H) histology type. Asterisks (***) indicate $p < 0.001$; ns indicates non-significant ($p > 0.05$).

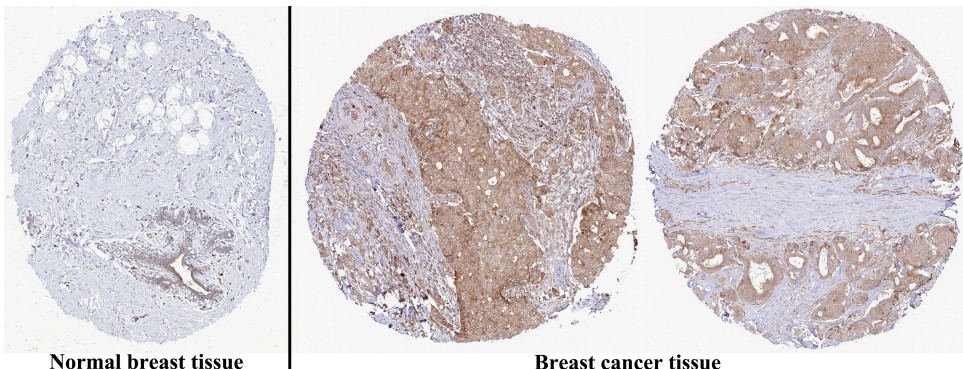

**Normal breast tissue**     **Breast cancer tissue**

**Figure 3** **Representative images of immunohistochemical staining of normal breast (left panels) and breast cancer (right panels, Lobular carcinoma (F), Duct carcinoma (R)) from Human Atlas.** Tissue slides were incubated with anti-PIMREG (1:150; HPA043783; Sigma Aldrich, St. Louis, MO, USA) at 4 °C, and the PIMREG protein level was increased in breast cancer tissue.

biological properties and prognoses, a subgroup analysis based on PAM50 subtypes was also performed, but did not indicate statistical significance (Fig. 6).

To further explore and identify the risk factors in patients with BC, a univariate Cox proportional-hazards regression analysis was conducted. The results showed that T stage, M stage, N stage, pathologic stage, age, and menopause status were associated with worse OS ($P = 0.046$, $P < 0.001$, $P < 0.001$, $P = 0.003$, $P < 0.001$, and $P = 0.003$, respectively)

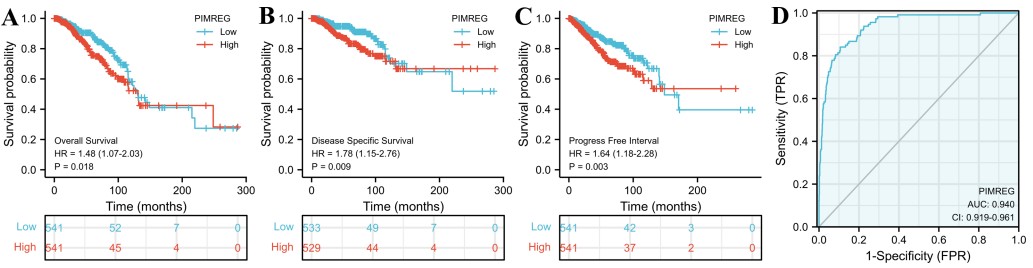

**Figure 4** **The outcomes of breast cancer in high PIMREG expression and low PIMREG expression groups (sample size = 1109 from the TCGA database).** (A) Overall survival (OS). (B) Disease-specific survival (DSS). (C) Progression-free interval (PFI). (D) The receiver operating characteristic curve (ROC).

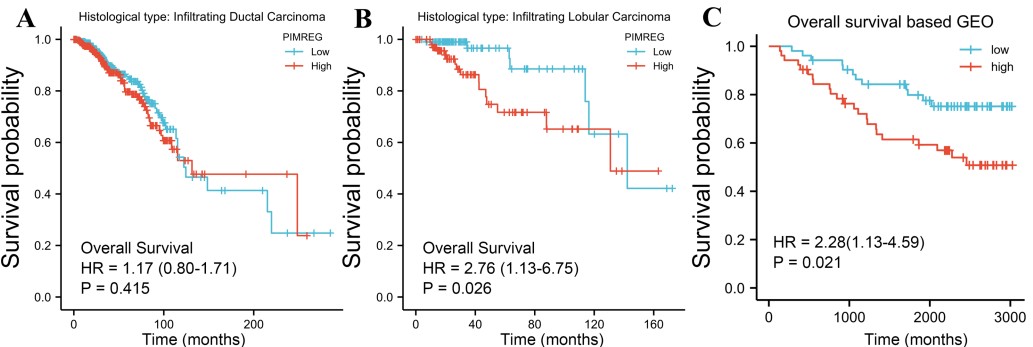

**Figure 5** **Subgroup overall survival (OS) analysis based on the low and high expression of PIMREG (sample size = 1109 from the TCGA database, 104 from the GEO database).** (A) Infiltrating ductal carcinoma. (B) Infiltrating lobular carcinoma. (C) OS based on the GEO database (GSE42568).

(Fig. 7). In addition, the higher expression of PIMREG was detected (HR = 1.475, 95%CI [1.070–2.034], $P = 0.018$) and was determined to be a significant predictor of worse OS.

The above results suggested that the T stage, N stage, M stage, age, pathologic stage, and menopause status might be linked to prognosis. As the pathologic stage of breast cancer is based on the TNM stage system, thus only the TNM stage was applied here for nomogram development. Therefore, a prognostic nomogram based on TNM stage, age, menopause, and PIMREG expression level was developed to predict individual survival probability (Fig. 8A). The C index of the nomogram was 0.741(0.713−0.769), indicating its potential for the prediction of overall survival. The calibration curve of this prediction model showed that the established lines of 1-, 3- and 5-year survival matched the ideal line at a relatively high degree (Fig. 8B). Collectively, these results indicated that PIMREG may be a prognostic signature for BC.

## Identities in PIMREG-related signaling pathway identified by GSEA

GSEA was conducted to identify the differences in signaling pathways between low- and high-PIMREG expression datasets (Fig. 9). The result revealed that PIMREG was related to

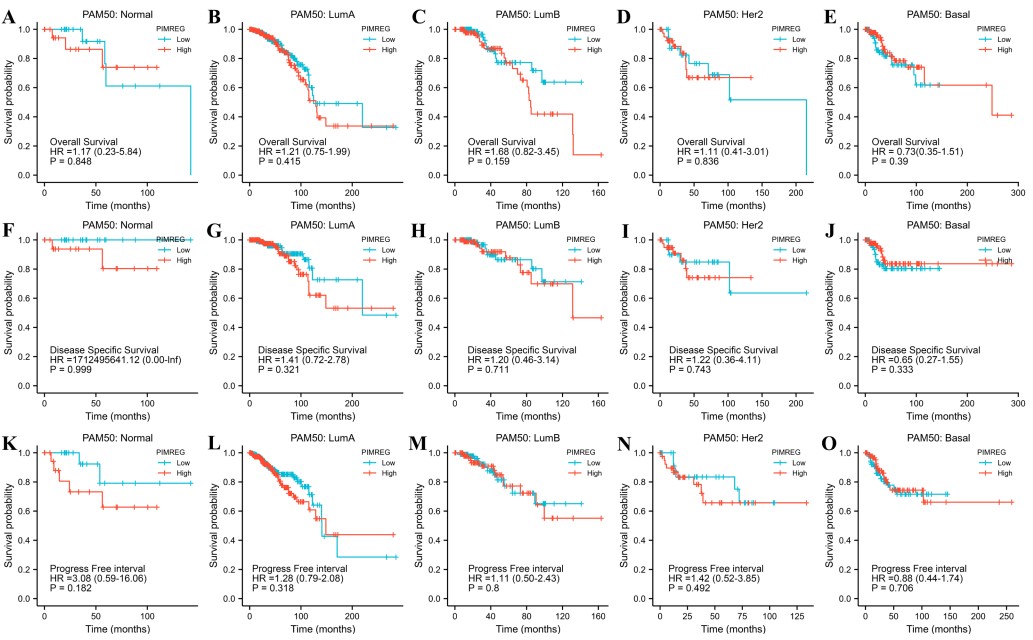

**Figure 6** **The outcomes of breast cancer in high PIMREG expression and low PIMREG expression groups based on PAM50 subtypes (sample size = 1109 from TCGA database).** (A) Overall survival (OS) of Normal-like subtype, (B) OS of Luminal A subtype, (C) OS of Luminal B subtype, (D) OS of HER2-enriched subtype, (E) OS of Basal-like subtype, (F) Disease-specific survival (DSS) of Normal-like subtype, (G) DSS of Luminal A subtype, (H) DSS of Luminal B subtype, (I) DSS of HER2-enriched subtype, (J) DSS of Basal-like subtype, (K) Progression-free interval (PFI) of Normal-like subtype, (L) PFI of Luminal A subtype, (M) PFI of Luminal B subtype, (N) PFI of HER2-enriched subtype, (O) PFI of Basal-like subtype.

the neuroactive ligand–receptor interaction, NABA-secreted factors, cell cycle checkpoint, Class A1 Rohdopsin-like receptor, DNA repair, mRNA splicing, and translation pathways.

## Immune cell infiltration of PIMREG in BC

The relationship between PIMREG and immune cell infiltration was studied using the TIMER database, as immune cell levels often correspond to cancer cell proliferation and progression (Fig. 10) (*Grenier, Yeung & Khanna, 2018*). A statistically significant correlation was found between PIMREG expression and the infiltration of macrophages, neutrophils, and dendritic cells. However, the correlation coefficients did not quite reach conventional levels of a positive Spearman's rho value (<0.3).

To further explore the association between PIMREG and immunomodulators, three immunoinhibitors that were most correlated with PIMREG were Lymphocyte Activating 3 (LAG3), Indoleamine 2,3-dioxygenase 1 (IDO1), and CTLA-4 (Figs. 11A–11C). On the other hand, the three immunostimulators that showed the strongest correlation with PIMREG were poliovirus receptor (PVR), human UL16-binding protein 1 (ULBP1), and tumor necrosis factor (TNF)-receptor superfamily 13C (TNFRSF13C) (Figs. 11D–11F).

Regarding major histocompatibility complex (MHC) molecules, transporter associated with antigen processing 2 (TAP2), transporter associated with antigen processing 1 (TAP1),

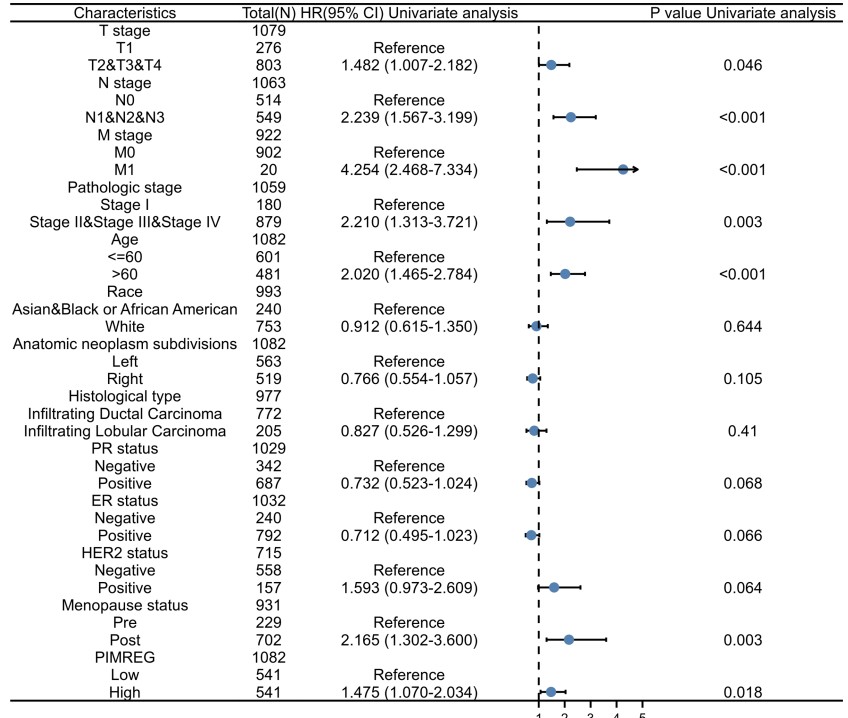

**Figure 7** **Forest plot of univariate analysis of the correlation between clinicopathological characteristics and overall survival in BRCA.** Reference as the control group, blue dots as the estimated HR value, and black lines as 95% confidence interval of HR, sample size = 1109.

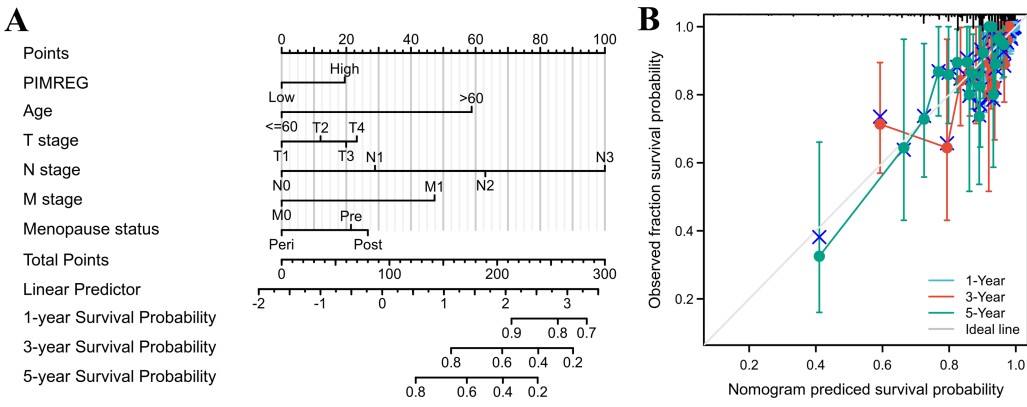

**Figure 8** **The predicting role of PIMREG regarding the prognosis of BRCA patients (sample size = 1109).** (A) Nomogram curve. The value of each variable was given a score on the point scale axis. The total score could be easily calculated by adding every score to predict the 1-, 3-, and 5-year survival probability (B) Calibration curve. The 45-degree line represents an ideal match between the actual survival ($Y$-axis) and nomogram-predicted survival ($X$-axis) and the perpendicular line means 95% confidence intervals.

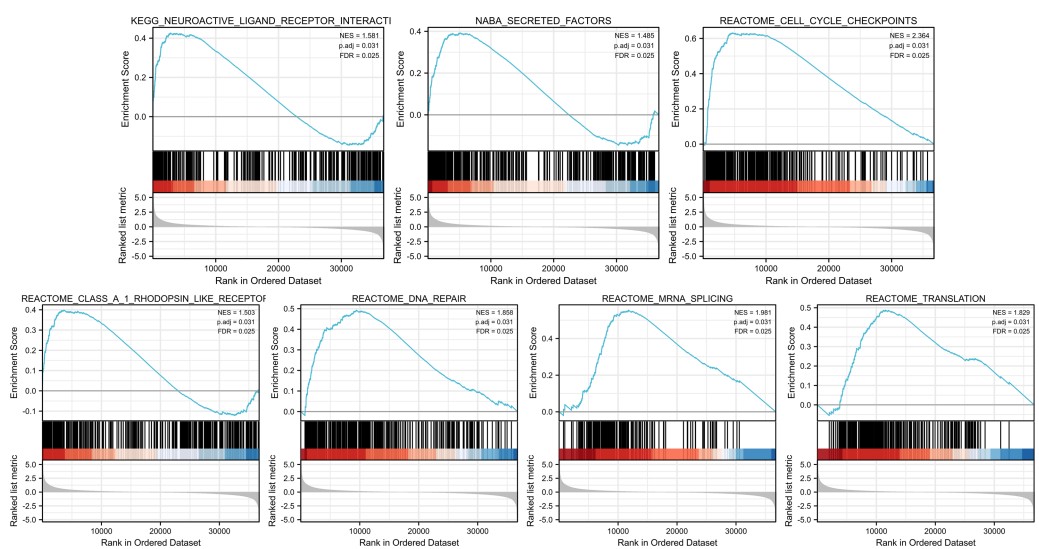

**Figure 9** GSEA web server results of the comparison between high PIMREG expression and low expression in breast cancer (sample size = 1109). NES: normalized enrichment score; p. adj; adjusted *p*-value; FDR: false discovery rate.

and human leukocyte antigen-A (HLA-A) showed significant associations with PIMREG (Figs. 11G–11I). These findings suggest that PIMREG may be involved in immune cell infiltration and immune modulation in breast cancer.

## Relationship between PIMREG and immune checkpoints in BC

The relationship between PIMREG and immune checkpoints, including PD1 (PDCD1), PD1-L1 (CD274), and CTLA-4, which play a crucial role in tumor immune evasion, was assessed (*Zhang et al., 2021*). According to the analysis of TIMER data, PIMREG expression showed a relatively strong positive correlation with CTLA-4 only in BRCA (invasive breast carcinoma). This correlation was further adjusted by tumor purity using partial Spearman's correlation (Fig. 12A). However, the analysis using GEPIA data did not find a significant positive correlation between PIMREG and PD1 (PDCD1), PD1-L1 (CD274), and CTLA-4 (Figs. 12B–12D).These findings suggest that the relationship between PIMREG and immune checkpoints may vary depending on the specific analysis method and dataset used. Further studies are needed to explore the potential interactions between PIMREG and immune checkpoints in breast cancer.

## Other genes/proteins interacted with PIMREG in patients with BC

GeneMANIA analysis revealed that PIMREG is implicated in various functions, including protein deacetylation, CHD-type complex, transcription repressor complex, histone deacetylase complex, ATPase complex, and SWI/SNF superfamily-type complex. The top 20 interactors associated with PIMREG included phosphatidylinositol binding clathrin assembly protein (PICALM), Tudor domain-containing 7 (TDRD7), RB binding protein 7 (RBBP7), RB binding protein 4 (RBBP4), metastasis-associated 1 family member 2 (MTA2), histone deacetylase 2 (HDAC2), suppressor APC domain-containing 2 (SAPCD2), kinesin

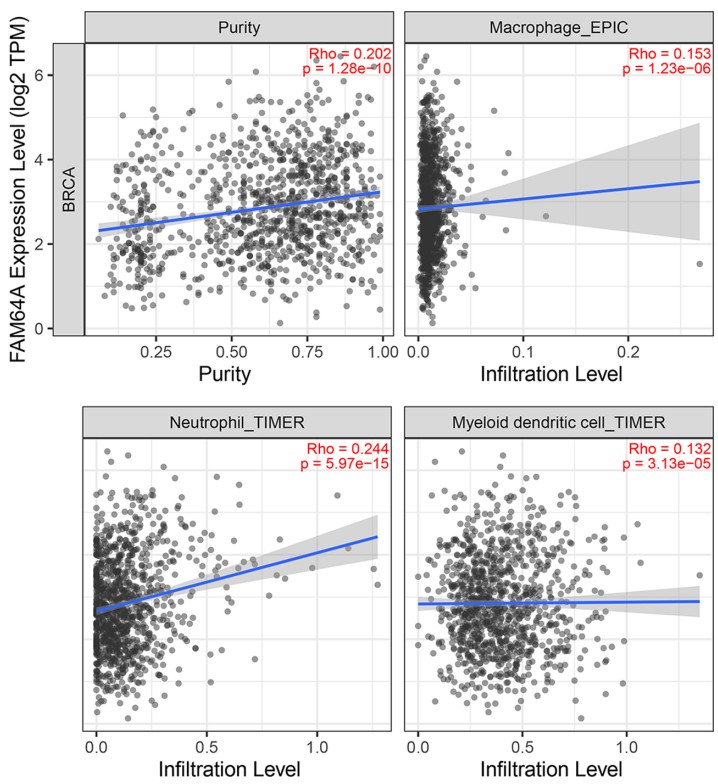

**Figure 10** **The association between levels of immune infiltrating cells and PIMREG expression in breast cancer.** Sample size = 1109, dots represent gene expression , blue lines represent linear association and dark grey areas represent the 95% confidence intervals.

family member 15 (KIF15), chromatin licensing and DNA replication factor 1 (CDT1), histone deacetylase 1 (HDAC1), prion protein (PRNP), PHD finger protein 19 (PHF19), thymidylate synthetase (TYMS), zyg-11 family member B (ZYG11B), cyclin E1 (CCNE1), BUB1 mitotic checkpoint serine/threonine kinase (BUB1), ubiquitin-associated and SH3 domain-containing B (UBASH3B), SPC24 component of NDC80 kinetochore complex (SPC24), BLM RecQ like helicase (BLM), and transforming acidic coiled-coil-containing protein 3 (TACC3) (Fig. 13A).

Network analysis using MaxLink of BRCA1, BRCA2, and PIMREG revealed that MTA2, Cell Division Cycle 20 (CDC20), HDAC2, and Cyclin-Dependent Kinase 1 (CDK1) were important links connecting PIMREG with BRCA1/2 (Fig. 13B). Among these links, HDAC2 was the only direct connection between PIMREG and both BRCA1 and BRCA2, and MTA2 and HDAC2 were identified as interactors *via* GeneMANIA.

## Investigate potential drugs that demonstrate interaction with PIMREG in BC

Using Q-omics (version 1.0), we analyzed 493 potential drugs. Among them, we identified 46 drugs that positively responded to increased expression of PIMREG (Fig. 14). The selection of these drugs was based on the Y-score, which represents the log (fold-change)

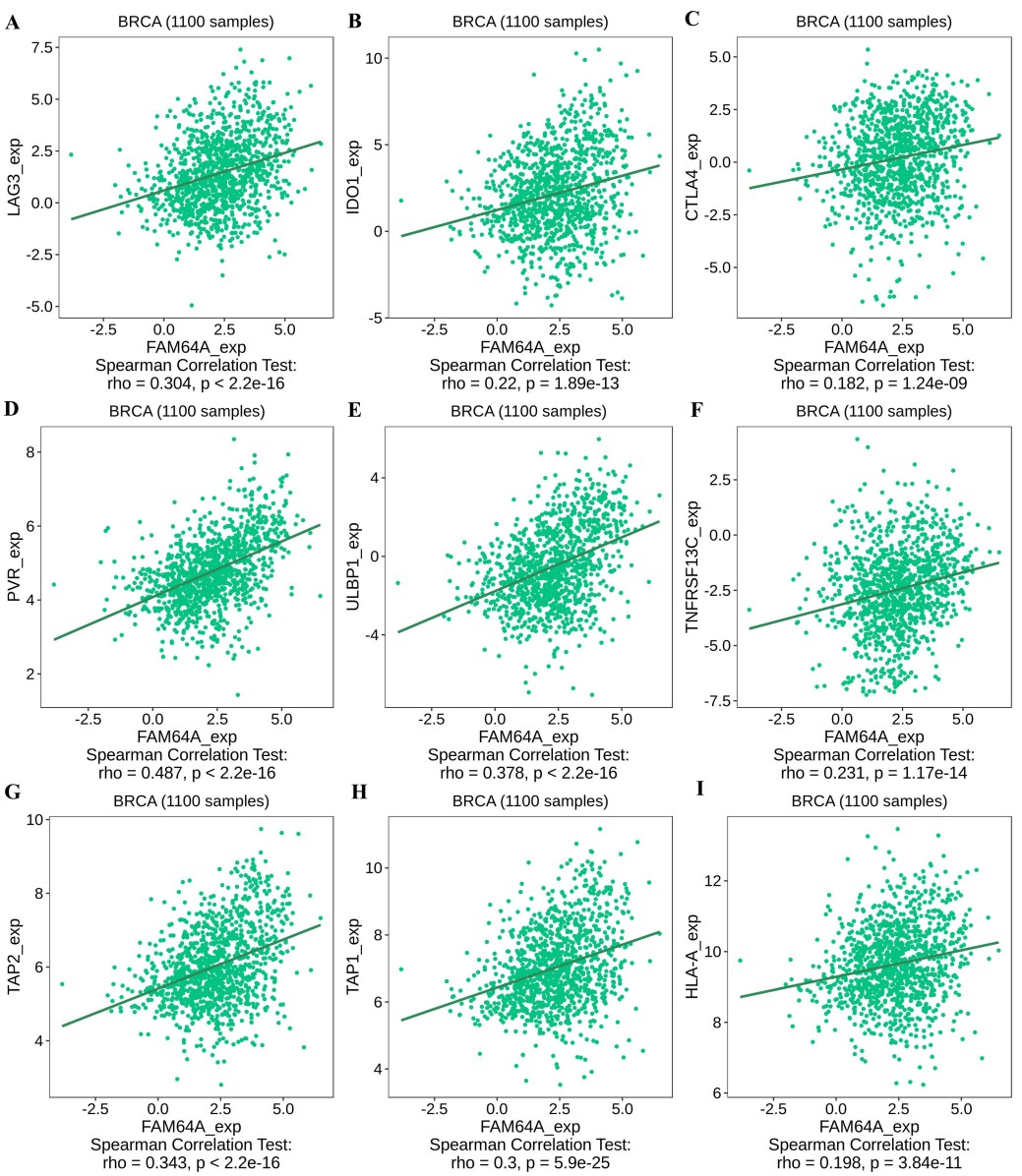

**Figure 11** **The correlation between the PIMREG expression level and immunomodulators in breast cancer based on TISIDB (sample size = 1100,dots represent gene expression and lines represent linear association).** (A–C) Immunomodulators that are highly correlated with PIMEG; (D–F) immunomodulators that are highly correlated with PIMREG; (G–I) MHC molecules that are highly correlated with PIMREG.

of the drug response between samples with high PIMREG expression and samples with low PIMREG expression. The top three responsive drugs, based on the Y-score, were Mitoxantrone, Bleomycin, and AZD7762. For more detailed information about these potential drugs, please refer to Supplementary File 1.

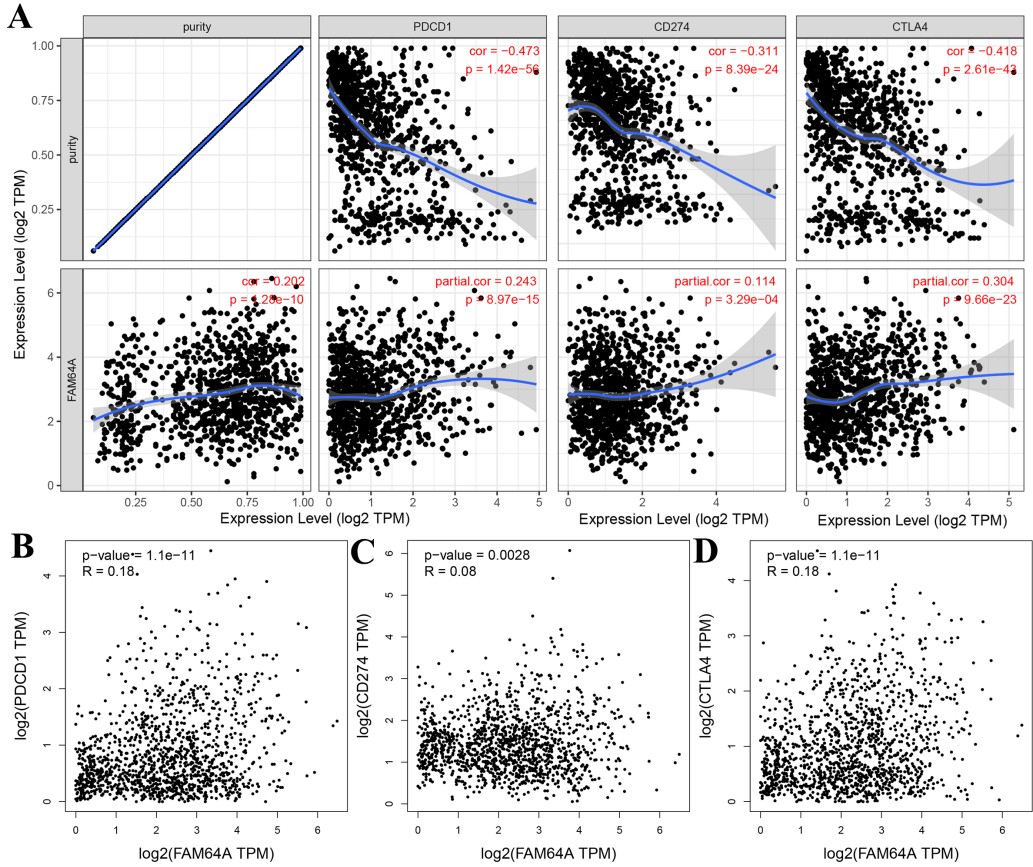

**Figure 12** **Correlation of PIMREG expression with PDCD1, CD274, and CTLA-4 expression in breast cancer (sample size = 1109, dots represent gene expression , blue lines represent linear association and dark grey areas represent the 95% confidence intervals).** (A) Spearman correlation of PIMREG with expression of PDCD1, CD274, and CTLA-4 in breast cancer. (B) The expression correlation of PIMREG with PDCD1 in breast cancer. (C) The expression correlation of PIMREG with CD274 in breast cancer. (D) The expression correlation of PIMREG with CTLA-4 in breast cancer.

## Knockdown of PIMREG decreased the cell proliferation and migration in MDA-MB-231 cells

As anticipated, both qRT-PCR and western blot assays revealed the overexpression of PIMREG in BC cells when compared to normal human mammary epithelial cells (Figs. 15A–15B, $P < 0.01$). The western blot assay conducted on tissue samples (Fig. 15C, $P < 0.01$) and the experimental IHC assay (Fig. 15D) further confirmed the overexpression of PIMREG in BC tissue samples. To assess the impact of PIMREG on cell proliferation and migration, siRNA was employed to knockdown PIMREG expression (Figs. 15E–15F, $P < 0.01$). Following transfection with PIMREG-siRNA, both the CCK8 assay and transwell migration assay demonstrated a significant reduction in cell proliferation and migration in MDA-MB-231 cells when compared to the untreated group (Figs. 15G–15H, $P < 0.01$). These findings suggest that PIMREG may play a crucial role in BC cell proliferation and migration. Additionally, the colony formation assay was conducted to verify the effects

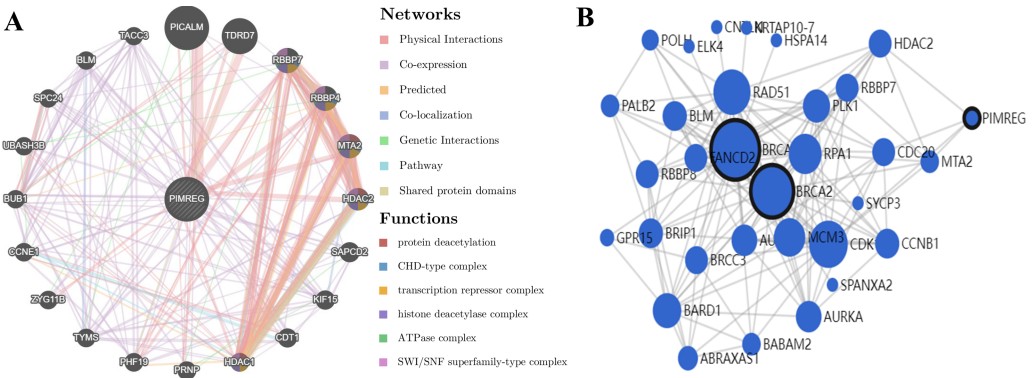

**Figure 13 PPI interactions:** (A) Interactions between other proteins and PIMREG in patients with breast cancer by GeneMANIA. (edge thickness represents interaction strength, color represents interaction type, node size represents protein score) (B) Network analysis of BRCA1, BRCA2, and PIMERG *via* MaxLink (node size represents the connectivity score).

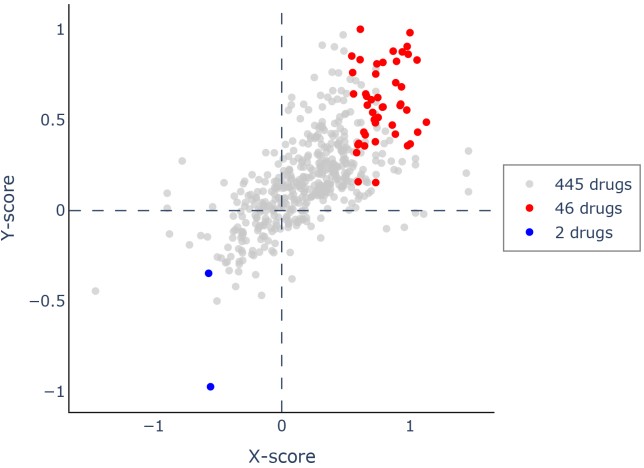

**Figure 14 Scatter plot of drug response to PIMREG *via* Q-omics.** Red dots and blue dots indicated the positive response and negative response, respectively.

of HDAC2 (Fig. 15I). The results demonstrated that the colony formation ability of MDA-MB-231 cells was reduced after the knockdown of HDAC2, indicating that HDAC2 may serve as a potential therapeutic target for breast cancer.

## DISCUSSION

BC is the most prevalent cancer type and a leading cause of tumor-associated mortality (15.5%) among females worldwide (*Sung et al., 2021*). Late-stage diagnosis, metastasis, and disease progression can result in irreversible consequences. However, timely detection and effective therapeutic interventions can significantly improve patient survival (*Barba et al., 2021*). Therefore, there is an urgent need to identify novel biomarkers that hold clinical and prognostic significance in BC. Additionally, the search for promising therapeutic targets

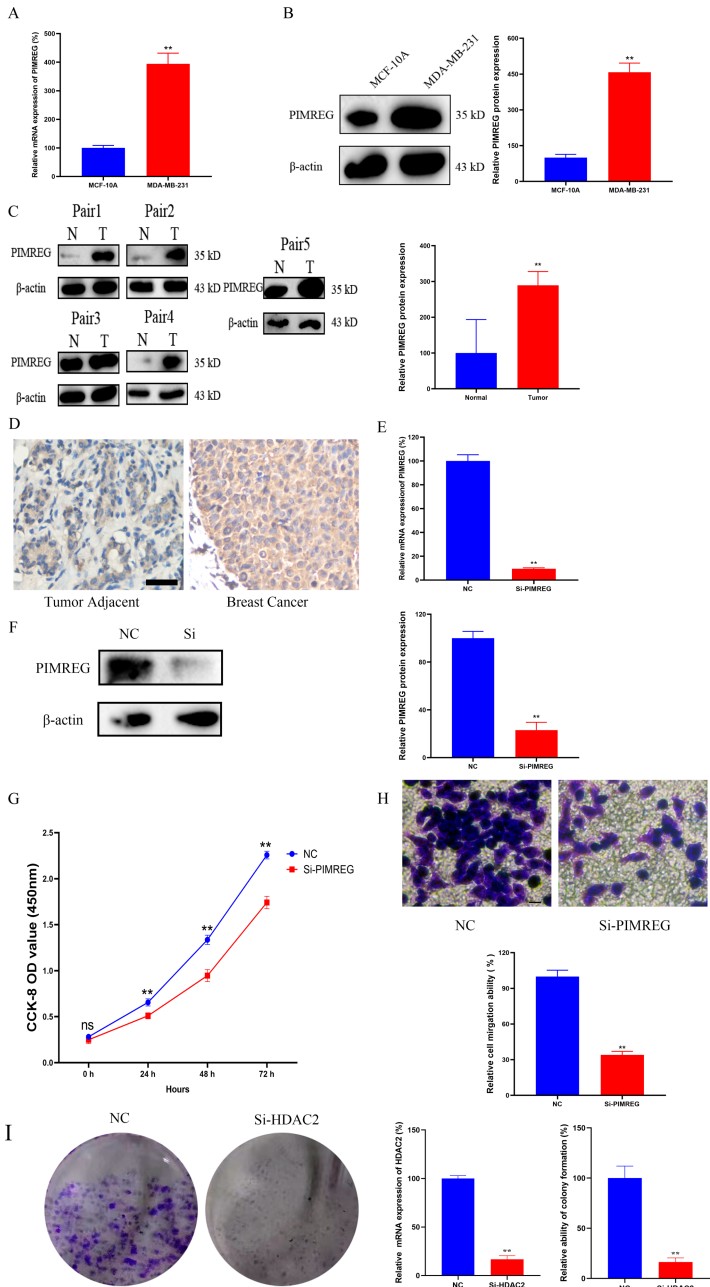

**Figure 15 Effects of PIMREG on BC cell proliferation and migration.** (A–B) mRNA and protein expression of PIMREG in MCF-10A and MDA-MB-231. (C) Protein expression of PIMREG in BC tissues and matched adjacent normal tissues. (D) IHC assay of PIMREG in BC tissues and matched adjacent normal tissues (Scar bar = 10 μm). (E–F) mRNA and protein expression of PIMREG in MDA-MB-231 cells transfected with siRNA- PIMREG. (G–H) cell proliferation and migration in MDA-MB-231 cells transfected with siRNA- PIMREG (Scar bar = 5 μm) colony formation ability in MDA-MB-231 cells transfected with siHDAC2. Data are presented as the mean ± standard deviation. ** $p < 0.01$ *versus* Normal group or NC group.

for anti-tumor drug development is crucial in order to enhance overall survival and delay tumor progression.

In this study, we conducted an assessment of the expression pattern of PIMREG across different subtypes of BC. Our findings align with previous literature, highlighting the crucial role of PIMREG in promoting the formation of breast cancer cells (*Yamada et al., 2018*). According to our study, bioinformatic analysis and western blot analysis showed that PIMREG expression levels were significantly higher than in normal tissues, which is consistent with the findings of the previous study (*Yao et al., 2019*). Furthermore, PIMREG showed high expression in advanced pathologic stages of BC. Additionally, receptor profiling revealed lower expression of PIMREG in estrogen receptor-positive (ER+) and progesterone receptor-positive (PR+) BC cases (*Yao et al., 2019*). In situations where accurate classification or controversy arises regarding ER or PR status, PIMREG could serve as a useful distinguishing marker to aid physicians in diagnosis. Histologically, PIMREG exhibited more substantial upregulation in infiltrating ductal carcinoma than infiltrating lobular carcinoma. Thus, the localization of this gene could be utilized as a supplemental biomarker to assist in the differential diagnosis between these subtypes.

We conducted further exploration of the impact of PIMREG overexpression on the prognostic outcome of BC. Our findings, based on a larger cohort of patients with diverse backgrounds, are consistent with a previous study indicating that overall survival (OS), disease-specific survival (DSS), and progression-free interval (PFI) are negatively associated with high expression of PIMREG (*Jiang et al., 2019*). The AUC value further confirms its good prognostic value, and analysis of data from the GEO database confirms its negative association with OS. Moreover, our *in vitro* study demonstrated that the knockdown of PIMREG in MDA-MB-231 cells decreased cell proliferation and migration. The poor prognosis associated with PIMREG overexpression may be attributed to excessive NF-$\kappa$B signaling, inhibition of the negative feedback loop, epithelial-to-mesenchymal transition (EMT), and enhanced breast cancer stemness (*Yao et al., 2019*; *Zhang et al., 2019*). Furthermore, PIMREG expression is negatively associated with survival in the infiltrating lobular carcinoma subtype. This pattern could potentially be explained by the influence of constitutive activation of AKT, which is a hallmark mutation of infiltrating lobular carcinoma. This activation amplifies downstream NF-$\kappa$B signaling, ultimately promoting tumor cell survival and proliferation (*Pramod et al., 2021*). However, it is important to note that the survival analysis based on PAM50 subtypes did not reach statistical significance, limiting its use in prognostic prediction. The limited sample size in each subgroup could contribute to this lack of significance, and future studies with larger sample sizes are needed to clarify the clinical significance of PIMREG in different PAM50 subtypes of BC.

An extensive evaluation of PIMREG outcomes, combined with univariate analysis and other clinically relevant patterns, was performed to develop a nomogram. The calibration plot demonstrated favorable consistency between the actual and predicted values for 1-, 3-, and 5-year survival. To provide personalized scores for individual patients, the model incorporated complementary perspectives specific to their respective tumors.

Consequently, our nomogram has the potential to serve as an invaluable prognostic tool for physicians.

To explore the mechanism underlying the prognostic value of PIMREG, GSEA was conducted. The results revealed several pathways and processes potentially implicated in its prognostic significance, including neuroactive ligand receptor interaction, Naba secreted factors, cell cycle checkpoints, Class A/1 rhodopsin-like receptors, DNA repair, mRNA repair, and translation. Notably, the association between PIMREG and Naba secreted factors, previously identified by *Naba et al. (2016)* highlights their relevance to extracellular matrix (ECM) proteins. ECM is known to be a critical component of metastatic niches and plays a pivotal role in breast cancer metastasis (*Hebert et al., 2020*). Furthermore, the involvement of cell cycle checkpoints suggests a potential role in promoting cell cycle arrest in breast cancer cells (*Akbarzadeh et al., 2022*; *Cao et al., 2022*; *Markowicz-Piasecka et al., 2022*).

Tumor growth and metastasis are highly dependent on the complex tumor microenvironment (TME) and the extracellular matrix (ECM) (*Hanahan & Coussens, 2012*; *Oliver et al., 2018*). The ECM serves as a multifaceted network that not only provides structural support but also supplies biochemical factors and imparts biomechanical cues crucial for tumorigenesis (*Buoncervello, Gabriele & Toschi, 2019*). Furthermore, an analysis of the relationship between PIMREG and tumor-infiltrating immune cells in breast cancer revealed positive correlations with macrophages, neutrophils, and dendritic cells (DCs). Previous studies have linked macrophage infiltration to poor prognosis in breast cancer (*Mehta et al., 2021*). Recent advancements have also highlighted the key role of neutrophils in breast cancer progression and metastasis (*Yang et al., 2021*). Moreover, DCs have been extensively investigated as potential targets in cancer therapy, and studies have shown that combining autologous dendritic cells with activated cytotoxic T cells may represent a novel approach for breast cancer treatment (*Shevchenko et al., 2020*).

The investigation of therapies targeting immunomodulators has also shed light on immune infiltrating cells. In line with our study findings, which indicate higher expression of PIMREG in ER-negative and PR-negative groups, this provides additional evidence of a potential positive association between PIMREG expression and LAG3, a negative regulator of T cells. Another immunoinhibitory pathway of interest is IDO1, a rate-limiting oxidoreductase (*Macchiarulo et al., 2009*), which is currently being explored as part of combined photodynamic therapy and checkpoint blockade immunotherapy for triple-negative breast cancer (*Wu et al., 2021*). Although CTLA-4 ranked third in terms of correlation with PIMREG, it only demonstrated a trend towards conventional positive correlation levels, and further studies on therapies targeting CTLA-4 in breast cancer are still necessary. Our study also identified immunostimulatory agents, such as ULBP1, which activates NK cells and exerts tumor-killing effects (*Qi et al., 2021*; *Wang et al., 2020*), and TNFRSF13C, which promotes cancer cell death (*Abo-Elfadl et al., 2020*; *Bhat et al., 2013*; *Véquaud et al., 2015*). The transporter associated with antigen processing (TAP) plays a critical role in peptide delivery, and cancer cells can modulate TAP1 and/or TAP2 levels to reduce peptide delivery, thereby evading recognition by cytotoxic CD8+ T cells (*El Hage, Durgeau & Mami-Chouaib, 2013*). HLA-A, a well-known major histocompatibility complex

(MHC) molecule, has been linked to prognosis in basal-like and HER2-enriched breast cancer subtypes (*Noblejas-López et al., 2019*).

In our study, we also investigated immune checkpoints, specifically PD-1, PD-L1, and CTLA-4, which are known to negatively regulate T-cell immune response. Activation of PD-1 by PD-L1 leads to a reduction in T-cell activity, cytokine production, and tolerance to antigens. Pembrolizumab, a PD-L1 inhibitor, has been tested in early phase clinical trials for metastatic PD-L1-positive triple-negative breast cancer (TNBC) and has shown clinical benefits in approximately 20% of patients (*Nanda et al., 2016*). However, our study did not reveal a significant correlation between PIMREG and PD-1/PD-L1, suggesting that PIMREG expression may not be a reliable biomarker for predicting the response to immunotherapies targeting PD-1/PD-L1. As for CTLA-4, previous studies have indicated that combining anti-CTLA-4 with anti-PD-1/PD-L1 may be a viable second-line therapy for BC (*Kern & Panis, 2021*; *Sun et al., 2020*). Our findings, based on TIMER analysis, suggested a correlation between PIMREG and CTLA-4, but further studies are needed to validate this correlation.

Apart from potential immunotherapies, exploration of other potential drugs that positively respond to PIMREG in BC was performed by using Via Q-omics. Mitoxantrone, a synthetic anthracycline anticancer drug, can produces antitumor effects by the inhibition of Toll-like receptor 4 (TLR4) and NF-$\kappa$B activation (*Evison et al., 2016*; *Rinne et al., 2020*). Mitoxantrone-based chemotherapy used to be a chemotherapeutic drug for BC decades ago (*Hainsworth et al., 1991*). Bleomycin is a commonly used medication for various malignancies, and it has recently gained favor as part of electrochemotherapy for cutaneous and subcutaneous metastases of BC (*Esposito et al., 2021*; *Ferioli et al., 2021*; *Russano et al., 2021*). AZD7762 is a selective checkpoint kinases 1 and 2 inhibitor, and a previous meta-analysis suggested an association between checkpoint kinase 2 mutation and increased risk of BC (*Liang et al., 2018*; *Ma et al., 2012*; *Park et al., 2016*). Therefore, AZD7762 is a promising drug in BC treatment which requires more evaluation.

To explore additional potential therapeutic targets, we investigated the relationship between PIMREG and BRCA1/2, as well as proteins highly associated with PIMREG. Although the specific mechanism and pathway interactions through which PIMREG promotes breast cancer remain unknown (*Archangelo et al., 2008*), our study identified correlations between PIMREG and MTA2, and HDAC2. MTA2 is known to deacetylate ER alpha and p53, inhibiting their transactivation function and potentially affecting breast cancer progression (*Cui et al., 2006*). HDAC2, a member of the histone deacetylase family, is located in the nucleus, and its overexpression has been associated with lymph node invasion, higher grade, and poor prognosis in breast cancer (*Darvishi et al., 2020*). Interestingly, our study revealed that HDAC2 is the only direct link between PIMREG and BRCA1 and BRCA2. BRCA1 and BRCA2 are well-known genetic factors involved in DNA repair and are associated with a strong predisposition for early-onset familial breast cancer (*Toh et al., 2008*). Furthermore, our colony formation assay results indicated that HDAC2 may serve as a potential therapeutic target for breast cancer. Based on the findings of our study and previous research, MTA2 and HDAC2 show promise as potential targets for breast cancer treatment.

However, this study has several limitations that should be considered. First, the datasets used for bioinformatic analysis were obtained from different databases and platforms, which may introduce uncertain systematic bias and potentially lead to misinterpretation of the data. Second, the enrichment analysis conducted to assess the regulatory function of PIMREG was preliminary and further experimental validation is necessary. Third, additional investigations are required to explore the interaction of potential drugs with PIMREG and their effects on breast cancer patients with high PIMREG expression. Fourth, the underlying mechanism of PIMREG in breast cancer remains largely unknown, and further *in vitro* and *in vivo* studies are needed to gain a deeper understanding.

## CONCLUSION

In summary, PIMREG is upregulated in BC tissues, and its high expression correlates with poor BC prognosis and immune infiltrates in BC. Among the proteins associated with PIMREG, HDAC2 might serve as a critical link between PIMREG and BRCA1/2, and thus could be a potential therapeutic target for future research. Potential immunomodulators, and immune checkpoints correlated with PIMREG were also identified in this study, indicating potential values of immunomodulatory drugs and immune checkpoint inhibitors in BC. Mitoxantrone, Bleomycin, and AZD7762 may respond well to PIMREG. Therefore, future studies are needed to further explore their therapeutic values.

## ACKNOWLEDGEMENTS

We thank for all the patients involved in this study. We acknowledge GEO, TCGA, and HPA databases for providing their platforms and data, and we thank for the contributors who uploaded their datasets for public access and use.

### Funding

This study was supported by grants from the National Natural Science Foundation of China (No. 82172190 to Ju Gao), the Jiangsu Association for Science and Technology Young Scientific and Technological Talents Support Project (No. 2021-008; Nanjing, China), the Jiangsu Province ''333'' High-level Talents Training Project (No. 2022-3-6-146), and the Yangzhou Science and Technology Plan Project (No. YZ2021148 to Ju Gao). Postgraduate Research & Practice Innovation Program of Jiangsu Province (No: KYCX22_2434, to Ruijin Xie). The funders had no role in study design, data collection and analysis, decision to publish, or preparation of the manuscript.

### Grant Disclosures

The following grant information was disclosed by the authors:
The National Natural Science Foundation of China: 82172190.
Jiangsu Association for Science and Technology Young Scientific and Technological Talents Support Project: 2021-008.

Jiangsu Province "333" High-level Talents Training Project: 2022-3-6-146.
Yangzhou Science and Technology Plan Project: YZ2021148.
Postgraduate Research & Practice Innovation Program of Jiangsu Province: KYCX22_2434.

## Competing Interests

The authors declare there are no competing interests.

## Author Contributions

- Wenjing Zhao performed the experiments, authored or reviewed drafts of the article, and approved the final draft.
- Yuanjin Chang performed the experiments, authored or reviewed drafts of the article, and approved the final draft.
- Zhaoye Wu performed the experiments, analyzed the data, authored or reviewed drafts of the article, and approved the final draft.
- Xiaofan Jiang performed the experiments, authored or reviewed drafts of the article, and approved the final draft.
- Yong Li performed the experiments, authored or reviewed drafts of the article, and approved the final draft.
- Ruijin Xie performed the experiments, analyzed the data, prepared figures and/or tables, authored or reviewed drafts of the article, and approved the final draft.
- Deyuan Fu analyzed the data, prepared figures and/or tables, and approved the final draft.
- Chenyu Sun conceived and designed the experiments, prepared figures and/or tables, and approved the final draft.
- Ju Gao conceived and designed the experiments, authored or reviewed drafts of the article, and approved the final draft.

## Human Ethics

The following information was supplied relating to ethical approvals (i.e., approving body and any reference numbers):

All the patients involved in this study signed a written informed consent and this study was approved by the ethics committee of Northern Jiangsu People's Hospital Affiliated to Yangzhou University (Approval ID: 2022KY316).

## Data Availability

The raw data, including original Western blots, are available in the Supplemental Files.

## Supplemental Information

Supplemental information for this article can be found online at http://dx.doi.org/10.7717/peerj.15703#supplemental-information.

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
