# Peer review of "Identification of PIMREG as a novel prognostic signature in breast cancer via integrated bioinformatics analysis and experimental validation"

_PeerJ, doi:10.7717/peerj.15703_

## Round 0.1 · original submission · Minor Revisions

Please address the concerns of all reviewers and amend the manuscript accordingly.

Reviewer 1 ·

Basic reporting

The Discussion part seems too long. For example, the paragraph between line 403 and 412, the authors didnot link the results from this study with the literature, thus it may be revised. Also, the paragraphs between line 413 and 444 are too long. Overall, the author should focus on the highlights of this study where the discussion should be conducted, and probably shortened.
There are several problems with the figures. First, the font in some Figures (for example, Fig 2, 5, 6, 9) is too small. Second, in figure 1 B, "p < 0.001" should be replaced with ***; In the figure 15, ratios of font in the figures should be unified, for example, the font in F is too bigger than that in C. In figure 3, the author should indicate how the PIMREG protein was stained in breast cancer tissue.

Experimental design

Majority of the results from this study are obtained by bio-informatic research. For the fundamental conclusion, the author may add experimental validation. For example, if direct evidence that supports the conclusion "HDAC2 may be a therapeutic target for breast cancer" can be added, this manuscript will be much better for publication.

Validity of the findings

no comment

Additional comments

1. line 13, add "(BC)" after "breast cancer";
2. line 22, the authors should explain what are "TIMER, TISIDB, and GEPIA";
3. line 51, FAM64A seems not to be the shortened name of "PICALM interacting mitotic regulator";
4. line 90, delete the 2nd "using";
5. line 213, the authors should explain what's the meaning of "5000 per well"
6. line 251, full name of HER2 should be put in this line, other than line 252;
7. line 415, is "LAG" in this line same with "LAG3" in line 417?
8. line 420, is "IDO1" in this line same with "IDO" in line 422?

Reviewer 2 ·

Basic reporting

Please see the Additional comments.

Experimental design

Please see the Additional comments.

Validity of the findings

Please see the Additional comments.

Additional comments

The authors combined bioinformatics and cell experiments to explore the role of PIMREG in breast cancer pathogenesis and prognosis. Overall, this study is suitable for publication, only if the authors address the following issues:

1. Throughout the manuscript, it seems better to use Grammarly (https://www.grammarly.com/) to check & correct potential grammatical errors. For example,
1.1 In DISCUSSION, it seems better to change "According to our study, bioinformatic analysis and western blot analysis shown that PIMREG expression levels were significantly higher than normal tissues, which is consistent with the findings of the previous study" into "According to our study, bioinformatic analysis and western blot analysis showed that PIMREG expression levels were significantly higher than normal tissues, which is consistent with the findings of the previous study".

2. In all FIGURES, it would be more clear and more readable to expand on figure legends by explaining the meanings of colors, groups, lines, and abbreviations. For example,
2.1 In Figure 1B and its legend, it would be more accurate to replace "Normal" (x-axis label) with "Tumor Adjacent".
2.2 In the legends of Figures 1–2, Figures 4–12, and Figure 15, it would be more rigorous to clearly state the sample size.
2.3 In Figure 3's legend, it would be better to clearly state that the images were cited from the Human Protein Atlas.
2.4 In Figure 5's legend, it would be more readable to clearly state that the figure is the "Subgroup overall survival (OS) analysis" based on the low and/or high expression of PIMREG.
2.5 In Figure 5C, it would be more rigorous to replace "Time" (x-axis label) with "Time (months)".
2.6 In Figure 7's legend, it would make the figure readable to explain the meaning of "Reference" and the blue dots as well as black lines.
2.7 In Figure 8' legend, please explain all elements in the figure to help readers understand the figure.
2.8 In Figure 9's legend, please rewrite its title, because the seven terms could not be all results of the "GSEA analyses". Please add modifiers to the title to limit it to a specific group (for example, "top 7"). However, the seven terms did not seem the top 7, so please explain why the authors selected these seven biological processes.
2.9 In Figure 10' legend, please explain all elements (especially the black dots, blue lines, and grey shadow) in the figure to help readers understand the figure.
2.10 In Figure 11' legend, please explain all elements (especially the green dots and lines) in the figure to help readers understand the figure.
2.11 In Figure 12's legend, please explain all elements (especially the black dots, blue lines, and grey shadow) in the figure to help readers understand the figure.
2.12 In Figure 13's legend, please explain all elements (especially the different sizes of nodes, distinct thickness of edges, and nodes highlighted in black circles) in the figure to help readers understand the figure.
2.13 In Figure 14's legend, it would help readers better understand the figure to explain who "the positive response and negative response" was targeted at.
2.14 In Figure 15, it would be more readable to label the two groups in Figure 15D and Figure 15H.

These revisions would greatly help readers, who do not specialize in bioinformatics, to understand the results and their implications easily and efficiently.

3. In ABSTRACT:
3.1 In Background, it seems better to change "This study aimed to explore the clinical significance of PIMREG in BC" into "This study aimed to explore the clinical significance of PIMREG in breast cancer (BC)", because it is recommended to use the full name when the abbreviation is first introduced.
3.2 In Methods ("The Cancer Genome Atlas (TCGA) and Gene Expression Omnibus (GEO) databases were used to gather relevant information"), it seems clearer to expand on what "relevant information" were gathered.
3.2 In Methods, it seems better to change "Genes/proteins that interact with PIMREG in BC were also identified through GeneMANIA and MaxLink" into "Genes/proteins that interact with PIMREG in BC were also predicted through GeneMANIA and MaxLink", which would be more accurate.
3.3 In Methods, it seems clearer to specify whose "Gene set enrichment analysis (GSEA) was then performed"; in other words, please what the "Gene set enrichment analysis (GSEA) was then performed" on.

4. In INTRODUCTION:
4.1 In Paragraph 1, please explain the logic flow between "While the prediction of clinical prognosis in BC is related to clinical, pathological, and molecular characteristics" and "the underlying etiology of BC aggressiveness is yet unknown". Otherwise, it seems better to change "While the prediction of clinical prognosis in BC is related to clinical, pathological, and molecular characteristics, the underlying etiology of BC aggressiveness is yet unknown" into "The prediction of clinical prognosis in BC largely depends on our understanding of the clinical, pathological, and molecular characteristics of BC, but the underlying etiology of BC aggressiveness is yet unknown", which seems more logical.
4.2 In Paragraph 1, it seems better to change "Hence, identifying potential diagnostic and prognostic biomarkers for detection is warranted" into "Identifying potential diagnostic and prognostic biomarkers of BC would help both detect BC patients and understand BC etiology", which seems more logical and cohesive.
4.3 In Paragraph 2, it seems better to change "PIMREG, also known as PICALM interacting mitotic regulator (FAM64A), was discovered to be a clathrin assembly lymphoid myeloid leukemia gene (CALM) interactor that is located at 17p13.2 that modulates the subcellular localization of leukemogenic fusion protein CALM/AF10" into "PIMREG, also known as PICALM interacting mitotic regulator (FAM64A), was discovered to be a clathrin assembly lymphoid myeloid leukemia gene (CALM) interactor, located at 17p13.2, which modulates the subcellular localization of leukemogenic fusion protein CALM/AF10", which would be clearer.
4.4 In Paragraph 2, it seems better to change "PIMREG has been shown to control the transition from metaphase to anaphase of the cell’s mitotic cycle, so it could be a marker for the proliferation and tumorigenesis of various cancer kinds", which would be clearer and more cohesive.
4.5 In Paragraph 2, it would be better to delete "However, little is known about its function and correlation with tumor immunity in BC", which seems a repetition of the sentences ("However, the underlying mechanism ...") after it.
4.6 In Paragraph 2, the authors highlighted a knowledge gap in "the immune infiltrates associated with the upregulation of PIMREG", so it seems more logical to add sentences introducing how immune responses have been reported to BOTH be important in breast cancer AND be associated with PIMREG, based on the literature. This revision would justify why the authors conducted the analysis about immune cells.
4.7 In Paragraph 3, it seems better to change "To further investigate the roles and clinical values of PIMREG in BC, we conducted a comprehensive bioinformatics analysis by combining information, technology, and molecular biology" into "In this study, we combined bioinformatics analysis and cell biology experiments to investigate the roles and clinical values of PIMREG in BC", which would be more accurate and specific. In addition, it seems better to delete "As accumulating information has become available for a variety of diseases, along with the application of new techniques and methods, such as next-generation sequencing (NGS) [1316], bioinformatics approaches are widely utilized to explore and investigate the mechanisms and pathogenesis of malignancies at the molecular level, as well as to search and identify potential diagnostic and prognostic biomarkers and to look for potential targets for new therapeutic molecules and drugs[17-19]", which did not seem to contribute a lot. Furthermore, it seems better to change "In this study, we first explored the prognostic and diagnostic significance of PIMREG with data extracted from the Cancer Genome Atlas (TCGA) database" into "We first explored the prognostic and diagnostic significance of PIMREG using data extracted from the Cancer Genome Atlas (TCGA) database".
4.8 In Paragraph 3, it seems better to change "We then uncovered the importance and underlying mechanism of PIMREG in BC through a comprehensive strategy, including Gene set enrichment analysis (GSEA) and network analysis" into "Moreover, we expored the role of PIMREG in BC pathogenesis through Gene set enrichment analysis (GSEA) and network analysis", which would be more accurate and rigorous.

5. In RESULTS:
5.1 It would be clearer to end each paragraph in RESULTS with one sentence: "Together, these results suggest that ..." (a pattern like PMID: 34715879, PMID: 34384362, PMID: 35965679, and PMID: 34537192), summarizing a paragraph AND highlighting the implications of all results in the paragraph.
5.2 In "Upregulation of PIMREG in BC" ("Further analysis found a significant difference between pathologic stage I and stage II"), it seems clearer to specify whether the PIMREG expression was higher or lower in the stage II.
5.3 In "Identities in PIMREG-related signaling pathway identified by GSEA", it would be clearer to expand on this section by adding more details about how the authors separated "low- and highPIMREG expression datasets", what was the input of GSEA, and why the authors postulateed that "The result" (of "low- and highPIMREG expression datasets") would reveal that PIMREG was related to ...". Likewise, it would be easier to understand to elaborate on the Figure 9's legend, whose current version seems elusive and not readable.
5.4 In "Immune cell infiltration of PIMREG in BC", it seems more rigorous to add references to support "the immune cell level frequently corresponds with the proliferation and progression of cancer cells".
5.5 In "Immune cell infiltration of PIMREG in BC" ("However, it did not quite reach conventional levels of positive correlation"), please explain what is the meaning of "not quite reach conventional levels of positive correlation", which seems unclear.
5.6 In "Relationship between PIMREG and immune checkpoints in BC", it seems better to add references about breast cancer to support "immune checkpoints PD1 (PDCD1), PD1-L1 (CD274), and CTLA-4 are critical in the immune escape of tumors", if the authors could find relevant studies.
5.7 In "Relationship between PIMREG and immune checkpoints in BC", it would be clearer to explain how the authors "adjusted by purity".
5.8 In "Other genes/proteins interacted with PIMREG in patients with BC", please justify why the authors predicted genes/proteins interacting with PIMREG. In other words, please explain how the results could contribute to the aim of this study.

6. In CONCLUSION, it seems more rigorous to change "Mitoxantrone, Bleomycin, and AZD7762 respond well to PIMREG" into "Mitoxantrone, Bleomycin, and AZD7762 may respond well to PIMREG".

7. In SUPPLEMENTAL FILES, it would be better to add "peerj-83227-raw_data-study_approval" that are in English, for the convenience of international readers.

·

Basic reporting

Include more references and the literature review should be expanded
“Breast cancer (BC) is the most commonly diagnosed malignant tumor among females around the globe, and it is the leading cause of cancer-related fatalities, accounting for 6.9% of all cancer diagnoses.” Please give exact references for this data.

Abbreviations should be clearly defined
For example, what PIMREG stands for,
Phosphatidylinositol binding clathrin assembly protein interacting mitotic regulator (PIMREG) is a protein associated with cell proliferation.

Check for typos throughout the document
CO2 at a 37°C incubator. _ there should be a space between the number and the unit and fix superscript and subscript
• In brief, 5 μm thickness - 5 μM -
fix these kind of minor errors in order to enhance the quality of writing.
• 5 × 104 cells were seeded in the 8-μM
• further in vivo

Experimental design

• Even though the authors have given the reference paper it is good to briefly explain the 2−∆∆Ct value method.
• Please define the terms GSEA, TIMER and GEPIA (Gene Expression Profiling Interactive Analysis) and make it clear to the reader. Especially it is helpful for readers outside the field to understand what kind of studies are these.

Validity of the findings

Figure 3 Representative images of immunohistochemical staining of samples from normal breast (left panels) and breast cancer (right panels)
• What is the difference between 2 images that come under the breast cancer tissue? Clearly define what those two images are and what the difference etc is.

Additional comments

In this study the authors obtained data from different databases for the purpose of their bioinformatic analysis, therefore there is a possibility to misinterpret data, however it can be considered as a limitation of the study.
The paper was well written and can enhance the quality of writing by fixing minor errors. Accept the paper with minor revisions.

---

## Round 0.2 · accepted · Accept

All concerns of the reviewers were addressed and the revised manuscript is acceptable now.